# DILATED TEMPORAL ATTENTION AND RANDOM FEATURE PARTITIONING FOR EFFICIENT FORECASTING

## ABSTRACT

For years, Transformers have achieved remarkable success in various domains such as language and image processing. Due to their capabilities to capture long-term relationships, they are expected to give potential benefits in multivariate long-term time-series forecasting. Recent works have proposed *segment*-based Transformers, where each token is represented by a group of consecutive observations rather than a single one. However, the quadratic complexity of self-attention leads to intractable costs under high granularity and large feature size. In response, we propose *Efficient Segment-based Sparse Transformer (**ESSformer**)*, which incorporates two sparse attention modules tailored for segment-based Transformers. To efficiently capture temporal dependencies, ESSformer utilizes Periodic Attention (PeriA), which learns interactions between periodically distant segments. Furthermore, inter-feature dependencies are captured via Random-Partition Attention (R-PartA) and ensembling, which leads to additional cost reduction. Our empirical studies on real-world datasets show that ESSformer surpasses the forecasting capabilities of various baselines while reducing the quadratic complexity.

## 1 INTRODUCTION

*Time-series forecasting* is a fundamental machine learning task that aims to predict future events based on past observations. A forecasting problem often requires long-term prediction and includes multiple variables: for example, stock price forecasting requires multiple market value predictions over a long temporal horizon. For this problem called *multivariate long-term time-series forecasting (M-LTSF)*, it is important to capture both (*i*) long-term temporal dependencies between past and future events and (*ii*) inter-feature dependencies among different variables. For decades, M-LTSF has been of great importance in various applications such as health care (Nguyen et al., 2021; Jones et al., 2009), meteorology (Sanhudo et al., 2021; Angryk et al., 2020), and finance (Qiu et al., 2020; Mehtab & Sen, 2021).

In recent years, there have been developed a number of deep neural architectures for M-LTSF problems, including linear models (Chen et al., 2023a; Zeng et al., 2022), state-space models (Rangapuram et al., 2018; Gu et al., 2022), and RNNs (Lin et al., 2023b; Du et al., 2021). Among them, Transformer-based methods have proliferated (Zhou et al., 2021; Liu et al., 2022b; Lim et al., 2020; Wu et al., 2022; Zhou et al., 2022; Li et al., 2020; Chen et al., 2023b; Zhao et al., 2023; Zhang et al., 2023; Shao et al., 2023; Yu et al., 2023; Lin et al., 2023a; Nie et al., 2023; Zhang & Yan, 2023) because of their intrinsic capabilities to capture long-term dependencies. In the realm of Transformers for M-LTSF, *segment-based* Transformers, which encode a group of consecutive observations (*i.e.*, a segment) into each token, have achieved state-of-the-art performance (Zhang & Yan, 2023; Nie et al., 2023). The excellence of segment-based Transformers comes from their tokenization which imbues each token with richer semantics, compared to conventional observation-based approaches where each token encodes a single observation.[1] However, when the number of segments is large because of fine-grained segments or high-dimensional features, segment-based Transformers suffer from high complexity which originates from the quadratic cost of self-attention.

**Contribution.** To address the inefficiency in the segment-based Transformers, we propose *Efficient Segment-based Sparse Transformer (**ESSformer**)* based on our two efficient self-attention modules:

---

[1]According to granularity for tokenization, we call them observation-based or segment-based Transformers.

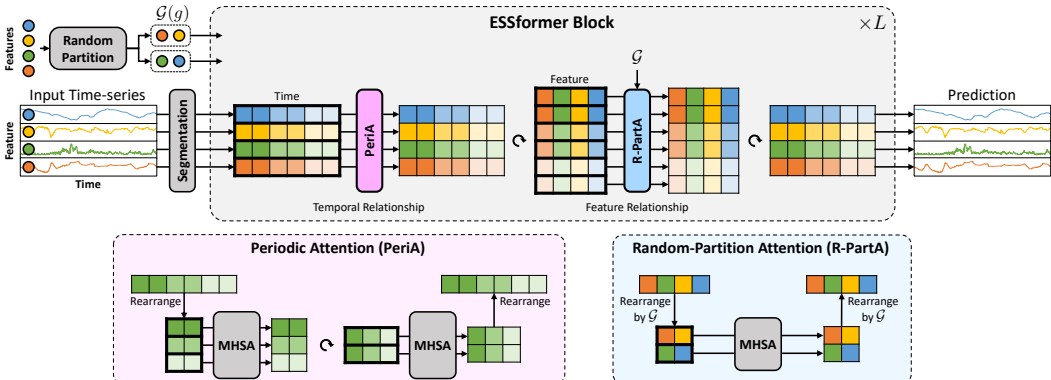

Figure 1: Architecture of Efficient Segment-based Sparse Transformer (ESSformer) with Periodic Attention (PeriA) and Random-Partition Attention (R-PartA) where 'MHSA' denotes vanilla multi-head self-attention in (Vaswani et al., 2017).

(*i*) *Periodic Attention (PeriA)* across the temporal dimension and (*ii*) *Random-Partition Attention (R-PartA)* across the feature dimension. To be specific, based on our observation that a periodic pattern appears in the self-attention matrix of segment-based Transformers, we design an efficient temporal attention module, Periodic Attention (PeriA), by composing a dilated attention with stride $P$ and a block-diagonal attention with a block size $P$. This leads to computational cost reduction in a temporal attention layer from $\mathcal{O}(N_S^2)$ to $\mathcal{O}(N_S^{1.5})$ given $N_S$ number of segments as input.[2] To capture dependencies between numerous features, we also design an efficient inter-feature attention module, Random-Partition Attention (R-PartA), by partitioning features randomly into groups of equal size $S_G$ and masking the attention matrix between different groups. This also reduces the attention cost from $\mathcal{O}(D^2)$ to $\mathcal{O}(DS_G)$ where $D$ is the feature size. During training, we find that inherent stochasticity in the random partitioning of R-PartA leads to efficient yet effective training. In the inference stage, we use an test-time ensemble technique to overcome the limitation that entire inter-feature relationships cannot be captured from this masked attention.

We conduct comprehensive experiments on a variety of benchmark datasets for M-LTSF. Across 27 out of 28 experimental scenarios, ESSformer surpasses the forecasting performance of 11 recent baselines achieving a 1.036 average rank (see Table 1). These baselines span a spectrum from Transformer-based (Zhang & Yan, 2023; Nie et al., 2023; Zhou et al., 2022; Liu et al., 2022b; Zhou et al., 2021; Chen et al., 2023b; Zhao et al., 2023; Xue et al., 2023; Gao et al., 2023; Lin et al., 2023a), linear-based (Chen et al., 2023a; Zeng et al., 2022), and convolution neural network-based (CNN) models (Wang et al., 2023; Wu et al., 2023) to implicit neural representations (INRs) (Woo et al., 2023). Also, we demonstrate that ESSformer achieves the most efficient computational complexity among various segment-based Transformers. Finally, we reveal the useful characteristic of ESSformer in a challenging real-world scenario, the robustness under missing inputs.

To sum up, our contributions are summarized as follows:

- We propose Efficient Segment-based Sparse Transformer (ESSformer) for multivariate long-term time-series forecasting (M-LTSF). To efficiently capture temporal dependencies even with a large number of segments, we design Periodic Attention (PeriA) that reduces the computational cost via approximation through periodically sparse attention (see Section 3.1).

- We also design Random-Partition Attention (R-PartA) that further reduces the computational cost of inter-feature attention under a large feature size, by randomly partitioning features into multiple groups and capturing only intra-group connections. For inference, we utilize an test-time ensemble method to capture complete inter-feature relationships (see Section 3.2).

- Our experimental results show the dual benefits of ESSformer equipped with PeriA and R-PartA: improving forecasting performance in M-LTSF, as well as enhancing efficiency. For instance, ESSformer achieves the best performance in 27 out of 28 tasks in M-LTSF (see Table 1).

---

[2]If $P = \sqrt{N_S}$, the computational cost of our temporal attention becomes $\mathcal{O}(N_S^{1.5})$.

## 2 PRELIMINARIES

### 2.1 MULTIVARIATE LONG-TERM TIME-SERIES FORECASTING (M-LTSF)

Before getting into the main topic, for ease of notation, let $[N : M]$ denote the set of integers between $N$ and $M$, where $N$ is inclusive and $M$ is exclusive (*i.e.*, $[N : M] := \{N, N + 1, \ldots, M - 1\}$). A $D$-variate time-series observation at time $t$ can be written as $\mathbf{x}_t = \{\mathbf{x}_{t,d} \in \mathbb{R} | d \in [0 : D]\} \in \mathbb{R}^D$, with $\mathbf{x}_{t,d}$ denoting the real-valued observation of the $d$-th feature at time $t$. The goal of time-series forecasting is to predict future observations $\{\mathbf{x}_t\}_{t \in [T, T+\tau]}$ based on previous observations $\{\mathbf{x}_t\}_{t \in [0, T]}$, with $T$ and $\tau$ indicating the length of past and future time steps, respectively. In this work, we consider a challenging case where $D > 1$ and $\tau \gg 1$, a setting also known as *multivariate long-term* time-series forecasting (M-LTSF).

### 2.2 SEGMENT-BASED TRANSFORMERS FOR M-LTSF

Leveraging the capability of Transformers in learning contextualized representations given a sequence of tokens via self-attention (Vaswani et al., 2017), many have proposed novel tokenization techniques and attention-variants towards solving M-LTSF (Wu et al., 2022; Lim et al., 2020; Chen et al., 2023b). While the most naïve approach would be to have each token represent an observation in a single time step (Zhou et al., 2021; Liu et al., 2022b), recent works have shifted towards segmenting time-series data across the temporal dimension and considering each token to embed observations within a fixed time span (Nie et al., 2023; Zhang & Yan, 2023), which intuitively leads to semantically richer input tokens (*e.g.*, the same stock price in a bull or bear market can have distinct meanings). Throughout the paper, we distinguish the two lines of research as *observation-based* and *segment-based* Transformers based on the granularity in tokenization. Our empirical study in Section 4 verifies that segment-based approaches indeed outperform observation-based counterparts in M-LTSF.

Prior to discussing our approach for segment-based M-LTSF, we finish the section by introducing several notations as well as our input tokenization procedure that follows previous work on M-LTSF (Nie et al., 2023; Zhang & Yan, 2023). Given multivariate time-series observations $\{\boldsymbol{x}_{t,d}\}_{t \in [0:T], d \in [0:D]}$, we temporally divide the sequence into $N_S$ segments of equal length.[3] In other words, the $b$-th segment of the $d$-th feature can be written as

$$\mathbf{s}_{b,d} = \left\{ \mathbf{x}_{t,d} \in \mathbb{R} | t \in \left[ \frac{bT}{N_S} : \frac{(b+1)T}{N_S} \right] \right\} \in \mathbb{R}^{\frac{T}{N_S}}. \tag{1}$$

Then, we pass through a linear layer to embed observations to latent space and add learnable temporal and feature-wise positional encodings, $\mathbf{E}^{\text{Time}} \in \mathbb{R}^{N_s \times d_h}$ and $\mathbf{E}^{\text{Feat}} \in \mathbb{R}^{D \times d_h}$:

$$\mathbf{H}_{b,d}^{(0)} = \texttt{Linear}(\mathbf{s}_{b,d}) + \mathbf{E}_b^{\text{Time}} + \mathbf{E}_d^{\text{Feat}} \in \mathbb{R}^{d_h}, \quad \mathbf{H}^{(0)} \in \mathbb{R}^{N_S \times D \times d_h}. \tag{2}$$

Given the initial representations $\mathbf{H}^{(0)}$ as input, a segment-based Transformer encoder with $L$ layers outputs the final representations $\mathbf{H}^{(L)}$, which is forwarded through a decoder to predict future observations. In this paper, we employ a linear-based decoder like Nie et al. (2023), where $\{\mathbf{H}_{b,d}^{(L)}\}_{b=1}^{N_S}$ are concatenated and mapped into future observations $\{\mathbf{x}_{t,d}\}_{t \in [T, T+\tau]}$ by a single linear layer.

## 3 METHOD

In this section, we introduce our novel framework for M-LTSF, **Efficient Segment-based Sparse Transformer (ESSformer)**, with two novel sparse attention mechanisms: *Periodic Attention (PeriA)* and *Random-Partition Attention (R-PartA)*, that efficiently learn temporal and inter-feature dependencies, respectively. Overall, given input segment representations $\mathbf{H}^{(0)}$ as described in Section 2.2, each layer of ESSformer is formulated as follows:

$$\bar{\mathbf{H}}^{(\ell-1)} = \mathbf{H}^{(\ell-1)} + \texttt{R-PartA}(\mathbf{H}^{(\ell-1)}, \texttt{PeriA}(\mathbf{H}^{(\ell-1)})), \tag{3}$$

$$\mathbf{H}^{(\ell)} = \bar{\mathbf{H}}^{(\ell-1)} + \texttt{MLP}(\bar{\mathbf{H}}^{(\ell-1)}), \quad \ell = 1, \ldots, L. \tag{4}$$

---

[3]In most scenarios, we can reasonably assume the input time span $T$ to be divisible by $N_S$ as we can adjust $T$ during data preprocessing as desired. In other cases when $T$ is not adjustable, we can pad with zeros as in Zhang & Yan (2023) and Wu et al. (2023).

In following Sections 3.1 and 3.2, we describe the details of our sparse attention mechanisms, periodic attention and random-partition attention, respectively. An overall illustration of our framework can be found in Figure 1.

## 3.1 PERIODIC ATTENTION (PERIA)

To first capture temporal relationships from input segments $\mathbf{H} \in \mathbb{R}^{N_s \times D \times d_h}$, PeriA processes the input through two attention modules, each uncovering distinct temporal relationships: (*i*) for *intra*-period relationships, a block-diagonal attention module with block size $P$ mixes features amongst segments within the same temporal period, and (*ii*) for *inter*-period relationships, a dilated attention with stride $P$ shares representations among periodically distant segments for longer-range contextualization.[4] Let $\mathtt{MHSA}(\mathbf{Q}, \mathbf{K}, \mathbf{V})$ denote the vanilla multi-head self-attention layer in Vaswani et al. (2017) where $\mathbf{Q}, \mathbf{K}$, and $\mathbf{V}$ are queries, keys and values. Also, when a collection of number $\mathcal{C}$ is given as index, it denotes to select all indices included in $\mathcal{C}$ (e.g., $\mathbf{H}_{\mathcal{C},d} = \{\mathbf{H}_{b,d}\}_{b \in \mathcal{C}} \in \mathbb{R}^{|\mathcal{C}| \times d_h}$). Then, the step-wise procedure of PeriA can be formulated as follows:

$$\forall i \in \left[0 : \frac{T}{P}\right], \quad \tilde{\mathbf{V}}^{\mathtt{PeriA}}(\mathbf{H})_{[iP:(i+1)P],d} = \mathtt{MHSA}(\mathbf{H}_{[iP:(i+1)P],d}, \mathbf{H}_{[iP:(i+1)P],d}, \mathbf{H}_{[iP:(i+1)P],d}),$$
(5)

$$\forall j \in [0 : P], \quad \mathtt{PeriA}(\mathbf{H})_{[j::P],d} = \mathtt{MHSA}(\mathbf{H}_{[j::P],d}, \mathbf{H}_{[j::P],d}, \tilde{\mathbf{V}}^{\mathtt{PeriA}}(\mathbf{H})_{[j::P],d}),$$
(6)

where $[j :: P]$ denotes the set of indices starting from $j$ with stride $P$ (*i.e.*, $[j :: P] := \{j, j + P, j + 2P, \dots\}$. After the block-diagonal attention captures intra-period relationships in equation 5, inter-period ones are considered in the dilated attention of equation 6.

**Why periodic attention?** Note that the cost of encoding $N_S$ segments through self-attention requires $\mathcal{O}(N_S^2)$ computational cost, which can be intractable when considering time-series data with large $T$. While enlarging the span of each segment can reduce $N_S$, previous works on Transformer-based generative modeling have shown that low segment granularity can deteriorate inference quality (Peebles & Xie, 2023; Jiang et al., 2021). Considering that time-series forecasting is analogous to generating future observations conditioned upon past signals (Rasul et al., 2021; Lim et al., 2023), this calls for an efficient architecture with sub-quadratic asymptotic cost in terms of the number of segments for practical deployment.

In light of such limitation, PeriA effectively imposes block diagonal and strided sparse attention masks, which reduces the computational cost without significantly sacrificing the expressivity of self-attention. Specifically, the periodically dilated sparsity structure is inspired based on an empirical observation shown in Figure 2, which depicts the attention score matrix of Crossformer after training on M-LTSF: the attention scores capturing temporal dependencies tend to mix representations of periodically spaced tokens. Additional visualizations from other segment-based Transformers can be found in Appendix I.2.

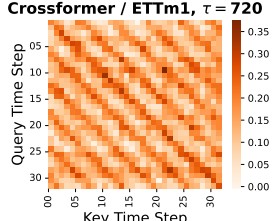

Figure 2: Periodic properties on attention maps.

In our experiments, we default to using period $P_* = 2^{\lceil log_2 \sqrt{N_S} \rceil} \approx \sqrt{N_S}$, which ultimately reduces the time and memory complexity from $\mathcal{O}(N_S^2)$ to $\mathcal{O}(N_S^{1.5})$. Our empirical results in Section 4 show that periodically sparse attention with $P_*$ is sufficient in maintaining the downstream capabilities of full attention. Furthermore, based on multi-periodicity in time series (Wu et al., 2023), we enlarge representation capabilities by changing $P$ in each layer (e.g., $P_1 = 2 \cdot P_*, P_2 = P_*, P_3 = \frac{1}{2} \cdot P_*$ for a 3-layer case). We further discuss the efficacy of our design choice in Appendix E.

## 3.2 RANDOM-PARTITION ATTENTION (R-PARTA)

With segment-based Transformers for M-LTSF, tokenizing each feature separately and modeling interactions among features in addition to temporal contextualization is known to boost downstream performance (Zhang & Yan, 2023). Unfortunately, this imposes an additional $\mathcal{O}(D^2)$ cost with full attention, which can be intractable when processing a large set of features $D$ similar to the previously discussed computational cost of encoding temporally long time-series data.

---

[4]We also assume $N_S$ to be divisible by $P$, which can easily be achieved by adjusting $T$.

To reduce the cost with respect to $D$, R-PartA first randomly partitions $D$ features into $N_G$ disjoint groups $\{\mathcal{G}(g)\}_{g \in [0:N_G]}$, with all groups having equal size $S_G$ (i.e., $\bigcap_{g \in [0:N_G]} \mathcal{G}(g) = \phi$, $\bigcup_{g \in [0:N_G]} \mathcal{G}(g) = [0:D]$, and $\forall g \in [0:N_G], |\mathcal{G}(g)| = S_G$).[5] A single partition is sampled prior to each forward step and is used throughout all layers of the model. Then, R-PartA mixes representations amongst features within the same group via block-diagonal attention:

$$\forall g \in [0:N_G], \quad \text{R-PartA}(\mathbf{H}, \mathbf{V})_{b,\mathcal{G}(g)} = \text{MHSA}(\mathbf{H}_{b,\mathcal{G}(g)}, \mathbf{H}_{b,\mathcal{G}(g)}, \mathbf{V}_{b,\mathcal{G}(g)}). \tag{7}$$

Note that this operation takes only intra-group interactions into account, thereby reducing the computational cost from $\mathcal{O}(D^2)$ to $\mathcal{O}(DS_G)$.[6] However, in the inference stage, if we run the forecasting procedure once, only partial inter-feature information within each group is considered. To overcome the limitation that entire information is not exploited, an test-time ensemble method involves running the forecasting procedure with random partitioning $N_E$ times and ensembling (i.e., averaging) $N_E$ forecasting outputs.

**Why does randomly partitioning features work?** In an experiment section (Section 4), we observe that this approach is helpful for not only efficiency but also forecasting performance. Our approach shows better forecasting performance than naïvely capturing dependencies among all features at once. We conjecture that this performance boost is due to R-PartA diversifying the training set by allowing partial feature interactions. This is equivalent to data augmentation (Gontijo-Lopes et al., 2021; Wen et al., 2021), as the partitioning process splits each multivariate time series into several concurrent time series with a subset of entire features, and its randomness leads to a different data sample at each iteration. As such, the random partitioning process generates diverse time series of sufficient dataset size from a small time-series dataset, leading to improvement in performance. In our later experiments (Figure 4), we find that the test-time performance of ESSformer is highly correlated with the number of possible partitions, which supports our speculation.

## 4 EXPERIMENTS

### 4.1 EXPERIMENTAL SETUP

We mainly follow the experiment protocol of previous segment-based Transformers (Zhang & Yan, 2023; Nie et al., 2023). A detailed description of datasets, baselines, and hyperparameters can be found in Appendix C.

**Datasets.** We evaluate ESSformer and other methods on the seven real-world datasets: (*i-iv*) ETTh1, ETTh2, ETTm1, and ETTm2 ($D = 7$), (*v*) Weather ($D = 21$), (*vi*) Electricity ($D = 321$), and (*vii*) Traffic ($D = 862$). For each dataset, four settings are constructed with different forecasting lengths $\tau$, which is in $\{96, 192, 336, 720\}$.

**Baselines**. We group existing baselines into five categories. First, Crossformer (Zhang & Yan, 2023) and PatchTST (Nie et al., 2023) are included in the segment-based Transformers. For observation-based Transformers, we use FEDformer (Zhou et al., 2022), Pyraformer (Liu et al., 2022b), and Informer (Zhou et al., 2021). We also compare against TSMixer (Chen et al., 2023a) and NLinear (Zeng et al., 2022) which are linear-based methods. We extend NLinear to a multi-variate version and denote it as NLinear-m. Refer to Appendix C.3 for how to modify NLinear into NLinear-m. Furthermore, we incorporate models based on CNNs and INRs: MICN (Wang et al., 2023), TimesNet (Wu et al., 2023), and DeepTime (Woo et al., 2023). Finally, we compare ESSformer against concurrent Transformer-based methods: JTFT (Chen et al., 2023b), GCformer (Zhao et al., 2023), CARD (Xue et al., 2023), Client (Gao et al., 2023), and PETformer (Lin et al., 2023a). According to code accessibility or fair experimental setting with ours, we select these concurrent models.

**Other settings.** ESSformer is trained with mean squared error (MSE) between ground truth and forecasting outputs. Also, we use MSE and mean absolute error (MAE) as evaluation metrics, and mainly report MSE. The MAE scores of experimental results are available in Appendix I.1. After training each method with five different random seeds, we measure the scores of evaluation metrics in each case and report the average scores.

---

[5]We assume that $D$ is divisible by $N_G$. We can overcome the cases where this divisibility doesn't hold by repeating or dropping some features. We elaborate on the solution in Appendix A.

[6]Our experiment (Figure 4(b)) shows that small $S_G$ is enough to attain good forecasting performance (e.g., $S_G = 20{\sim}30$ for 300${\sim}800$ features). Therefore, the empirical complexity is nearly linear.

Table 1: MSE scores of main forecasting results. OOM denotes out-of-memory in our GPU environments. The best score in each experimental setting is in boldface and the second best is underlined.

| Data | | Segment-based Transformer | | | Observation-based Transformer | | | Linear | | | Others | | |
|---|---|---|---|---|---|---|---|---|---|---|---|---|---|
| | | ESSformer | Crossformer | PatchTST | FEDformer | Pyraformer | Informer | TSMixer | NLinear | NLinear-m | MICN | TimesNet | DeepTime |
| ETTh1 | τ = 96 | **0.361** | 0.427 | 0.370 | 0.376 | 0.664 | 0.941 | **0.361** | 0.374 | 0.463 | 0.828 | 0.465 | 0.372 |
| | 192 | **0.396** | 0.537 | 0.413 | 0.423 | 0.790 | 1.007 | 0.404 | 0.408 | 0.535 | 0.765 | 0.493 | 0.405 |
| | 336 | **0.400** | 0.651 | 0.422 | 0.444 | 0.891 | 1.038 | 0.420 | 0.429 | 0.531 | 0.904 | 0.456 | 0.437 |
| | 720 | **0.412** | 0.664 | 0.447 | 0.469 | 0.963 | 1.144 | 0.463 | 0.440 | 0.558 | 1.192 | 0.533 | 0.477 |
| ETTh2 | 96 | **0.269** | 0.720 | 0.274 | 0.332 | 0.645 | 1.549 | 0.274 | 0.277 | 0.347 | 0.452 | 0.381 | 0.291 |
| | 192 | **0.323** | 1.121 | 0.341 | 0.407 | 0.788 | 3.792 | 0.339 | 0.344 | 0.425 | 0.554 | 0.416 | 0.403 |
| | 336 | **0.317** | 1.524 | 0.329 | 0.400 | 0.907 | 4.215 | 0.361 | 0.357 | 0.414 | 0.582 | 0.363 | 0.466 |
| | 720 | **0.370** | 3.106 | 0.379 | 0.412 | 0.963 | 3.656 | 0.445 | 0.394 | 0.460 | 0.869 | 0.371 | 0.576 |
| ETTm1 | 96 | **0.282** | 0.336 | 0.293 | 0.326 | 0.543 | 0.626 | 0.285 | 0.306 | 0.322 | 0.406 | 0.343 | 0.311 |
| | 192 | **0.325** | 0.387 | 0.333 | 0.365 | 0.557 | 0.725 | 0.327 | 0.349 | 0.365 | 0.500 | 0.381 | 0.339 |
| | 336 | **0.352** | 0.431 | 0.369 | 0.392 | 0.754 | 1.005 | 0.356 | 0.375 | 0.392 | 0.580 | 0.436 | 0.366 |
| | 720 | 0.401 | 0.555 | 0.416 | 0.446 | 0.908 | 1.133 | 0.419 | 0.433 | 0.445 | 0.607 | 0.527 | **0.400** |
| ETTm2 | 96 | **0.160** | 0.338 | 0.166 | 0.180 | 0.435 | 0.355 | 0.163 | 0.167 | 0.191 | 0.238 | 0.218 | 0.165 |
| | 192 | **0.213** | 0.567 | 0.223 | 0.252 | 0.730 | 0.595 | 0.216 | 0.221 | 0.260 | 0.302 | 0.282 | 0.222 |
| | 336 | **0.262** | 1.050 | 0.274 | 0.324 | 1.201 | 1.270 | 0.268 | 0.274 | 0.330 | 0.447 | 0.378 | 0.278 |
| | 720 | **0.336** | 2.049 | 0.361 | 0.410 | 3.625 | 3.001 | 0.420 | 0.368 | 0.416 | 0.549 | 0.444 | 0.369 |
| Weather | 96 | **0.142** | 0.150 | 0.149 | 0.238 | 0.896 | 0.354 | 0.145 | 0.182 | 0.162 | 0.188 | 0.179 | 0.169 |
| | 192 | **0.185** | 0.200 | 0.194 | 0.275 | 0.622 | 0.419 | 0.191 | 0.225 | 0.213 | 0.231 | 0.230 | 0.211 |
| | 336 | **0.235** | 0.263 | 0.245 | 0.339 | 0.739 | 0.583 | 0.242 | 0.271 | 0.267 | 0.280 | 0.276 | 0.255 |
| | 720 | **0.305** | 0.310 | 0.314 | 0.389 | 1.004 | 0.916 | 0.320 | 0.338 | 0.343 | 0.358 | 0.347 | 0.318 |
| Electricity | 96 | **0.125** | 0.135 | 0.129 | 0.186 | 0.386 | 0.304 | 0.131 | 0.141 | OOM | 0.177 | 0.186 | 0.139 |
| | 192 | **0.142** | 0.158 | 0.147 | 0.197 | 0.386 | 0.327 | 0.151 | 0.154 | OOM | 0.195 | 0.208 | 0.154 |
| | 336 | **0.154** | 0.177 | 0.163 | 0.213 | 0.378 | 0.333 | 0.161 | 0.171 | OOM | 0.213 | 0.210 | 0.169 |
| | 720 | **0.176** | 0.222 | 0.197 | 0.233 | 0.376 | 0.351 | 0.197 | 0.210 | OOM | 0.204 | 0.231 | 0.201 |
| Traffic | 96 | **0.345** | 0.481 | 0.360 | 0.576 | 2.085 | 0.733 | 0.376 | 0.410 | OOM | 0.489 | 0.599 | 0.401 |
| | 192 | **0.370** | 0.509 | 0.379 | 0.610 | 0.867 | 0.777 | 0.397 | 0.423 | OOM | 0.493 | 0.612 | 0.413 |
| | 336 | **0.385** | 0.534 | 0.392 | 0.608 | 0.869 | 0.776 | 0.413 | 0.435 | OOM | 0.496 | 0.618 | 0.425 |
| | 720 | **0.426** | 0.585 | 0.432 | 0.621 | 0.881 | 0.827 | 0.444 | 0.464 | OOM | 0.520 | 0.654 | 0.462 |
| Avg. Rank | | **1.036** | 7.214 | 2.857 | 6.929 | 10.286 | 10.429 | 2.786 | 4.607 | N/A | 8.000 | 7.25 | 4.357 |

Table 2: Test MSE of ESSformer compared to concurrent Transformer-based models for M-LTSF.

| Method | ETTm2 | | | | Weather | | | | Electricity | | | | Traffic | | | | Avg. Rank |
|---|---|---|---|---|---|---|---|---|---|---|---|---|---|---|---|---|---|
| | τ = 96 | 192 | 336 | 720 | 96 | 192 | 336 | 720 | 96 | 192 | 336 | 720 | 96 | 192 | 336 | 720 | |
| ESSformer | 0.160 | **0.213** | **0.263** | **0.337** | **0.142** | **0.185** | **0.235** | **0.305** | **0.125** | **0.142** | **0.154** | **0.176** | 0.345 | 0.370 | 0.385 | **0.426** | **1.250** |
| JTFT | 0.164 | 0.219 | 0.272 | 0.353 | 0.144 | 0.186 | 0.237 | 0.307 | 0.131 | 0.144 | 0.159 | 0.186 | 0.353 | 0.372 | **0.383** | 0.427 | 2.938 |
| GCformer | 0.163 | 0.217 | 0.268 | 0.351 | 0.145 | 0.187 | 0.244 | 0.311 | 0.132 | 0.152 | 0.168 | 0.214 | 0.377 | 0.393 | 0.414 | 0.445 | 4.563 |
| CARD | **0.159** | 0.214 | 0.266 | 0.379 | 0.145 | 0.187 | 0.238 | 0.308 | 0.129 | 0.154 | 0.161 | 0.185 | **0.341** | **0.367** | 0.388 | 0.427 | 2.813 |
| Client | 0.167 | 0.220 | 0.268 | 0.356 | 0.153 | 0.195 | 0.246 | 0.314 | 0.131 | 0.153 | 0.170 | 0.200 | 0.368 | 0.388 | 0.405 | 0.442 | 5.188 |
| PETformer | 0.160 | 0.217 | 0.274 | 0.345 | 0.146 | 0.190 | 0.241 | 0.314 | 0.128 | 0.144 | 0.159 | 0.195 | 0.357 | 0.376 | 0.392 | 0.430 | 3.625 |

## 4.2 FORECASTING RESULTS

Table 1 shows the test MSE of representative baselines along with the ESSformer. ESSformer outperforms baselines in 27 out of 28 tasks and achieves second place in the remaining one. It is worth noting that other segment-based Transformer baselines that do not take into account sparsity (e.g., Crossformer, PatchTST) underperform linear-based methods (e.g., TSMixer). We also provide visualizations of forecasting results in three segment-based Transformers in Appendix I.2: ESSformer, Crossformer, and PatchTST. In these figures, ESSformer catches temporal dynamics better than others. On top of that, ESSformer is compared to the five concurrent Transformer-based methods in Table 2. We use five datasets because the scores of the other cases are not available in JTFT and GCformer. ESSformer shows top-1 performance in 12 cases and top-2 in 16 cases out of 16 cases, attaining a 1.25 average rank.

## 4.3 ANALYSIS

While ESSformer aims to reduce the computational complexity of segment-based Transformers, forecasting performance is also improved in general. In this section, we conduct a plethora of ablation studies and hyperparameter analyses to better understand these results. We further test the

Table 3: Ablation study for PeriA and R-PartA by replacing them with vanilla multi-haed full self-attention (MHSA). Univariate ESSformer is the case where attention for inter-feature dependencies is removed (N/A), not considering any connection between features.

| ESSformer Variants | PeriA | R-PartA | ETTh2 ($D=7$) $\tau=96$ | 192 | 336 | 720 | Weather ($D=21$) 96 | 192 | 336 | 720 | Electricity ($D=321$) 96 | 192 | 336 | 720 | Traffic ($D=862$) 96 | 192 | 336 | 720 |
|---|---|---|---|---|---|---|---|---|---|---|---|---|---|---|---|---|---|---|
| Original (Multivariate) | PeriA | R-PartA | **0.269** | **0.323** | **0.317** | **0.370** | 0.142 | **0.185** | **0.235** | **0.305** | **0.125** | **0.142** | **0.154** | **0.176** | **0.345** | **0.370** | **0.385** | **0.426** |
| Ablated (Multivariate) | MHSA | R-PartA | 0.273 | 0.328 | 0.319 | 0.373 | 0.142 | **0.185** | 0.236 | **0.305** | **0.125** | 0.144 | 0.156 | 0.178 | 0.348 | 0.373 | 0.391 | 0.430 |
| | PeriA | MHSA | **0.269** | 0.325 | 0.318 | 0.371 | 0.146 | 0.192 | 0.244 | 0.307 | 0.129 | 0.147 | 0.163 | 0.204 | 0.363 | 0.383 | 0.394 | 0.441 |
| Ablated (Univariate) | PeriA | N/A | 0.272 | 0.325 | 0.318 | 0.374 | **0.141** | 0.186 | 0.237 | 0.308 | 0.128 | 0.146 | 0.163 | 0.204 | 0.368 | 0.388 | 0.404 | 0.441 |

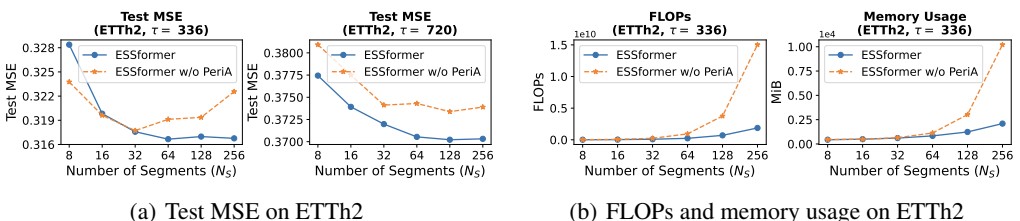

(a) Test MSE on ETTh2  (b) FLOPs and memory usage on ETTh2

Figure 3: The effect of PeriA on test MSE, the number of floating point operations (FLOPs), and memory usage by changing $N_S$.

robustness of ESSformer under challenging real-world scenarios where some features are missing. In this section, we only include the main experimental results so refer the readers to Appendix I for the additional experimental results or visualizations. Furthermore, we additionally explore the effect of changing $P$ per layer in Appendix E.

**Effectiveness of PeriA on forecasting performance.** We analyze the effect of PeriA during training. In Table 3, we substitute the PeriA with a full self-attention layer and compare its result against the original ESSformer. Due to the inductive bias inherent in a periodic form of attention, ESS-former with PeriA improves over its counterpart. Furthermore, in Figure 3(a), the influence of PeriA on test MSE is amplified when there are more segments.

**Effectiveness of R-PartA on forecasting performance.** Similar to the case of PeriA, we conduct an ablation study on R-PartA during training, in Table 3. Furthermore, we additionally explore the univariate case where R-PartA is removed so any connection between features is not considered. In both experiments, the original ESSformer with R-PartA outperforms all ablated cases. One interesting point is that the improvement of performance by R-PartA tends to be large, when the number of features is relatively large (e.g., in Electricity and Traffic). Because applying random partitioning to a dataset of larger features results in a larger dataset size of more diversity, we think that the amplified effect of R-PartA in Electricity and Traffic originates from more enhanced cardinality and diversity in datasets like data augmentation.

To further explore the effect of increased dataset sizes, we conduct two experiments in Figure 4 by adjusting $N_P$, which is the number of all possible partition choices that can be generated by random partitioning. In other words, $N_P$ can be regarded as the size of datasets. Firstly, we directly adjust $N_P$ and report test MSE of each case, in Figure 4(a). While $\{\mathcal{G}_g\}_{g\in[0,N_G]}$ is sampled from pools with all possible combinations in random partitioning, we instead limit the size of the partitioning pool into $N_P$ during training. 'Max' denotes the maximum size that the partitioning pool can have and original ESSformer is trained with $N_P = \text{Max}$. Secondly, the other way is to change $N_P$ indirectly by adjusting $S_G$, in Figure 4(b). The dotted line in this figure shows $N_P$ is changed by $S_G$. In both figures, when $N_P$ is large enough, ESSformer tends to show good forecasting performance. Therefore, from these results, we can infer that enhanced dataset size and diversity by the random partitioning might positively influence forecasting results.

At last, we examine the effect of the test-time ensemble method which is introduced to overcome the limitation that R-PartA cannot capture entire inter-feature relationships. In Figure 5, after training ESSformer with R-PartA, we use two types of attention for inter-feature dependencies in the inference stage: full attention with entire features and R-PartA with ensembling $N_E$ times. When

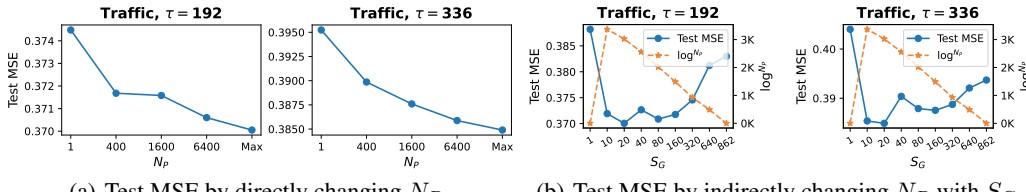

(a) Test MSE by directly changing $N_P$      (b) Test MSE by indirectly changing $N_P$ with $S_G$

Figure 4: Sensitivity to $N_P$ in Traffic. Note that according to $S_G$, $N_P$ is changed.

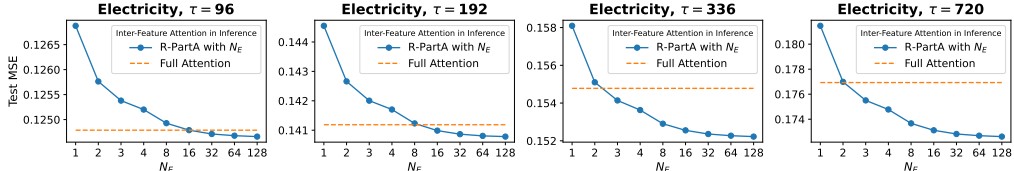

Figure 5: Sensitivity to $N_E$ in Electricity. Note that ESSformer for both lines are commonly trained with R-PartA but have different inter-feature attention in the inference stage.

$N_E$ is small, capturing entire dependencies between features has better performance than the other. However, when $N_E$ is sufficiently large, R-PartA with ensembling has similar performance to the case capturing entire information and sometimes even outperforms it. Therefore, with the test-time ensemble method and sufficient large $N_E$, ESSformer can address the constraint of not being able to capture complete inter-feature relationships. Note that we set $N_E$ to 3 for our main forecasting experiments, considering both efficiency and forecasting capabilities.

**Robustness of ESSformer under feature dropping.** In the real world, some features in multivariate time series are often missing. Inspired by the works that address irregular time series where observations at some time steps (Che et al., 2016; Kidger et al., 2020) are missing, we randomly drop some features of input time series in the inference stage and measure the increasing rate of test MSE in undropped features. For comparison, we use the original ESSformer and ablated one of which R-PartA is replaced with full attention featured in Table 3. ESSformer deals with the situation where some features are missing by simply excluding missing features in the random partitioning process, while the other with full attention has no choice but to pad dropped features with zeros. In Figure 7, unlike the ablated case, original ESSformer maintains its forecasting performance, regardless of the drop rate of the features. This robust characteristic gives ESSformer more applicability in real-world situations where some features are not available.

## 4.4 COMPLEXITY ANALYSIS

The theoretical complexity of two types of attention in various segment-based Transformers is compared in Table 4. ESSformer achieves the most efficient computation complexity in both attention cases. For temporal attention, because we set $P$ to $P_* \approx \sqrt{N_S}$, it achieves $\mathcal{O}(N_S^{1.5})$ complexity, which is the most efficient under fine-grained segments. Figure 3(b) shows that FLOPs and memory usage of ESSformer with and without PeriA whose complexities are $\mathcal{O}(N_S^{1.5})$ and $\mathcal{O}(N_S^2)$, respectively. These figures verify the empirical effectiveness of PeriA. Also, for inter-feature cases, Figure 4(b) shows that

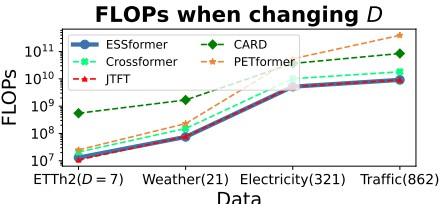

Figure 6: FLOPs of self-attention for inter-feature dependencies in various segment-based Transformers when changing $D$.

small $S_G$ is enough to generate sufficient and diverse datasets (e.g., $S_G = 20{\sim}30$ for $300{\sim}800$ features). As a result, ESSformer achieves the lowest or the second lowest FLOPs compared to Crossformer (Zhang & Yan, 2023), JTFT (Chen et al., 2023b), PETformer (Lin et al., 2023a), and CARD (Xue et al., 2023), as shown in Figure 6. While some existing approaches try to cheaply approximate dependencies among entire features with low-rank approximation, we employ the simpler yet enough effective approach of random feature partitioning.

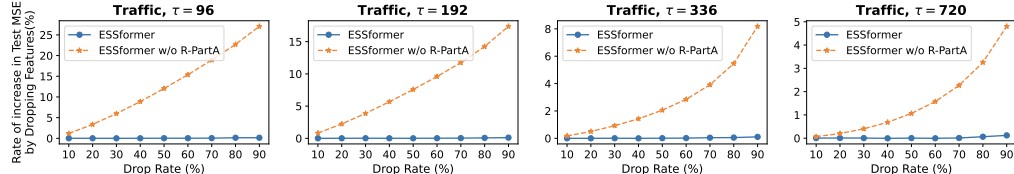

Figure 7: Increasing rate of test MSE by dropping $n\%$ features in ESSformer with or without R-PartA.

Table 4: Complexity comparison of two types of self-attention in segment-based Transformers: self-attention for temporal dependencies (Temp.) and for inter-feature dependencies (Feat.). For Temp. (resp. Feat.), we ignore $D$ (resp. $N_S$), because all the complexities are proportional to it.

| Type | ESSformer | Crossformer | PatchTST | JTFT | CARD | PETformer |
|---|---|---|---|---|---|---|
| Temp. | $\mathcal{O}(N_S^{1.5})$ | $\mathcal{O}(N_S^2)$ | $\mathcal{O}(N_S^2)$ | $\mathcal{O}(N_S^2)$ | $\mathcal{O}(N_S^2)$ | $\mathcal{O}(N_S^2)$ |
| Feat. | $\mathcal{O}(D)$ | $\mathcal{O}(D)$ | N/A | $\mathcal{O}(D)$ | $\mathcal{O}(D)$ | $\mathcal{O}(D^2)$ |

## 5 RELATED WORK

To capture temporal and inter-feature dependencies well, various approaches have been explored for time-series forecasting (Liu et al., 2022a; Bai et al., 2020; Liu et al., 2022c; Kollovieh et al., 2023; Shen & Kwok, 2023; Naour et al., 2023). However, because segment-based Transformers are mainly addressed in our work, we provide a brief explanation of existing segment-based Transformers. The first works that proposed the segment-based tokenization are Crossformer (Zhang & Yan, 2023) and PatchTST (Nie et al., 2023). Crossformer captures both temporal and inter-feature dependencies between segments, while PatchTST doesn't consider any relationships between features. Chen et al. (2023b) proposed a joint time-frequency domain Transformer (JTFT) which utilizes representation on frequency domains with learnable frequencies as well as one on time domains. Also, Channel Aligned Robust Dual Transformer (CARD) (Xue et al., 2023) introduces a robust loss function for time series forecasting to alleviate the potential overfitting issue with a new transformer architecture that considers 3 types of connections: relationships across temporal, feature, and hidden dimensions. Finally, Transformer with Placeholder Enhancement Technique (PETformer) (Lin et al., 2023a) proposed encoder-only structures, unlike other approaches which have encoder-decoder architectures.

Prior to finishing this section, because our work also focuses on sparse attention, we introduce some approaches to sparse attention for observation-based Transformers. Informer (Zhou et al., 2021) reduces complexity by selecting some valuable queries via the estimation of KL divergence between query-key distribution and uniform distribution. Pyraformer (Liu et al., 2022b) compute attention between nodes with different resolution to address both long-term and short-term temporal dependencies. Fedformer (Zhou et al., 2022) utilizes sparsity in the frequency domain by keeping the constrained number of frequency components. Note that these methods (*i*) do not achieve competitive performance on M-LTSF and (*ii*) only take account of the temporal dimension, unlike segment-based Transformers.

## 6 CONCLUSION

In segment-based Transformers, quadratic costs become inefficient under the high granularity of segments and a large feature size. To tackle the intractable costs, we propose ESSformer with PeriA and R-PartA which employ dilated temporal attention and random feature partitioning, respectively. In PeriA, based on observations about a periodical form of attention maps in segment-based Transformers, intra-period and inter-period relationships between different time steps are considered. As for R-PartA, we attain efficiency by randomly partitioning all features into multiple groups and capturing dependencies within each group. During inference, an test-time ensemble method enables R-PartA to consider entire relationships. Extensive experiments demonstrate the efficacy of ESSformer with two sparse attention modules in term of both forecasting performance and efficiency. Finally, we reveal the useful characteristics of our ESSformer in a challenging real-world scenario: robust forecasting under incomplete inputs.

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
