# OpenReview forum: "Periodic and Random Sparsity for Multivariate Long-Term Time-Series Forecasting"
_ICLR.cc/2024/Conference — Submitted to ICLR 2024_

### Official Review · Reviewer_h291 · 2023-10-28

**Soundness:** 4 excellent
**Presentation:** 4 excellent
**Contribution:** 3 good
**Rating:** 6
**Confidence:** 5

**Summary:**

The paper studies the problem of time-series forecasting using transformer models. The paper tries to improve the efficiency of the existing segment-based transformer model by two aspects: 1) the paper proposes the periodic attention block where data points in the same segments are first processed using self-attention and then only the data points with the same periodic indexed are processed together using a second self-attention. 2) The paper uses a random attention block on the feature where the features are firstly randomly partitioned into blocks and self-attention is applied accordingly based on the partitioning structure. The paper shows improved performance on the majority of time series forecasting task with improved efficiency.

**Strengths:**

The paper proposes two kinds of novel attention block mechanisms which are very intuitive and have strong empirical performance.

**Weaknesses:**

1. It seems that to make the periodic attention work, the model needs to know the right periodic interval of the time series. In some cases, this could be easy as we know the data is collected hourly/daily/etc but in the more general setting it may require to infer the actual periodic behaviors of the data stream. Moreover, there could be multiple and potentially overlapping periodic intervals encoded in the data stream. Extending the given approach to such a more general setting seems not trivial.

2. More studies on why the random partition attention works are necessary. It's not clear to decide on the number of groups and how many instances are required for the ensemble during inference to get the best performance. Are there some metrics we can check before/during training to get the right number of groups? Furthermore, how do they relate? I.e., if we change the number of groups when partitioning, how should the number of instances in the ensemble be adjusted accordingly?

**Questions:**

Please see the weaknesses part.

---

> ### Author Response · Authors · 2023-11-17
> **Response to Reviewer h291**
>
> Dear Reviewer h291,
>
> We sincerely thank you for your helpful feedback and insightful comments, especially for acknowledging the novelty and the strong empirical performance of ESSformer. In what follows, we address your concerns one by one.
>
> - - -
> **[Q1]** ESSformer needs to know the right periodic interval of the time-series data. \
> **[A1]** We agree that prior knowledge about the right periodic interval is important for better forecasting. However, it is hard to identify the right period as you pointed out that “there could be multiple and potentially overlapping periodic intervals encoded in the data stream”. Therefore, it is important to design an architecture that can implicitly capture various periodic patterns without the knowledge of the right period.
>
> To consider the above scenario and maintain efficiency simultaneously, as we mentioned in our original manuscript (the last paragraph in Section 3.1), we already used different periods across layers to enlarge representation capabilities. As reported in Appendix E, we found that it is better to use multiple periods (i.e., $[\frac{1}{2}P_{\*}, P_{\*}, 2P_{\*}]$) than a single period (i.e., $[P_{\*}, P_{\*}, P_{\*}]$). We think the improvement shows the potential of our architecture to be applied into a more general scenario.
>
> For your information, we provide the table in Appendix E showing the effect of multiple periods in the Traffic dataset:
>
> |Method                     |$\tau=$96|192  |336  |720  |
> |:-------------------------|:-------:|:---:|:---:|:---:|
> |$[\frac{1}{2}P_{\*}, P_{\*}, 2P_{\*}]$|**0.345**    |**0.370**|**0.385**|**0.426**|
> |$[P_{\*}, P_{\*}, P_{\*}]$    |0.346    |0.372|0.387|**0.426**|
>
> - - -
> **[Q2]** More studies on why the random partition attention works are necessary. Are there some metrics we can check before/during training to get the right number of groups and the number of instances for ensemble? How are they related? \
> **[A2]** Thank you for the interesting suggestion. Unlike the practices of the other domains (e.g., images), model selection without validation data is not a well-researched area in time series forecasting. Furthermore, given various properties in the benchmark datasets (e.g., dependency among features, originating domain, and number of features), finding a global rule for optimal hyperparameters without any validation data is a challenging task. Hence, following the convention in the forecasting literature, we simply choose the R-PartA hyperparameters, the group size $S_G$ and the number of instances for ensemble $N_E$, using the validation set.
>
> Along the way we conduct additional experiments to analyze the behavior of R-PartA, we find three empirical rules of thumb which can give help to selecting $S_G$ and $N_E$ before training. We here explain them and will include these analyses in Appendix H.
>
> **A large number of distinct partitions tends to perform well.** As shown in Figure 4(b) in the original manuscript, we found that it is better to select the size of each group in R-PartA, $S_G$, as what makes the available number of distinct partitions $N_{P}$ large because ESSformer benefits from the diversity of partitioning during training and inference. Note that $N_P$ is formulated by $S_G$ as follows:
> $$
> N_P = \prod_{i=0}^{\lfloor \frac{D}{S_G} \rfloor} {n - iS_G \choose S_G}/ \lfloor \frac{D}{S_G} \rfloor!.
> $$
> In this rebuttal period, we further extend this experiment to four datasets (ETTm1, Weather,
> Traffic, Electricity) where all datasets differ in feature size and originate from different domains. The experimental results (all the figures in Figure 9 of Appendix H) well verify that $S_G$ leading to large $N_P$ tends to perform well.
>
>
> **A large number of instances for ensembling tends to give better results.** As we have already shown in Figure 5 and Figure 8, the larger the number of instances for ensembling ($N_E$) is, the better ESSformer performs. Therefore, to achieve better forecasting results, large $N_E$ is required.
>
> **The effect of $N_E$ tends to be smaller, as $S_G$ increases.** We think that because ESSformer with small $S_G$ can capture relationships within very small groups, it might need large $N_E$ to recover the entire relationships from small ones. To prove this statement, we additionally conduct the experiments in Figure 10 of Appendix H where we measure performance gain of increasing $N_E$ in various $S_G$. This figure shows that the effect of $N_E$ tends to be smaller, as $S_G$ increases, supporting our statement. Therefore, this fact helps to select efficient but effective $N_E$ given $S_G$.

---

> ### Author Response · Authors · 2023-11-22
> **A Gentle Reminder to Reviewer h291**
>
> Dear Reviewer h291,
>
> Thanks again for your time and effort in reviewing our paper! As the discussion period is coming to a close, we would like to know if we have resolved your concerns expressed in the original reviews. We remain open to any further feedback and are committed to making additional improvements if needed. If you find that these concerns have been resolved, we would be grateful if you would consider reflecting this in your rating of our paper : )
>
> Best regards,
> Authors.

---

### Official Review · Reviewer_AR89 · 2023-10-31

**Soundness:** 3 good
**Presentation:** 3 good
**Contribution:** 2 fair
**Rating:** 3
**Confidence:** 5

**Summary:**

The authors propose a new time series forecasting method called Efficient Segment-base Sparse Transformer with two attention modules called PeriA and R-PartA. PeriA reduces the computational cost by taking advantage of the attention score matrix capturing temporal dependencies that tend to mix representations of periodically spaced tokens. R-PartA randomly partitions the given features into equal size groups and it seems to work well.

**Strengths:**

1. This article is exquisitely crafted, using precise and appropriate language while being rich in meticulous details, allowing readers to gain a deep understanding of the content. In terms of exposition, the article is both comprehensive and succinct, making it easily comprehensible without feeling overly lengthy or cluttered. Furthermore, the article also incorporates beautiful illustrations that are not only aesthetically pleasing but also vividly complement and enrich the content.
2. The article extensively experimented with the proposed model structure across seven datasets and outperformed the selected baseline.
3. The article not only conducted a theoretical analysis of the model's complexity but also presented a comparison of FLOPs and memory usage.

**Weaknesses:**

1. The effectiveness of the proposed two sparse attention modules in the model can be questionable. In the Method section (Section 3), I found that ESSformer is not a purely Transformer-based model because it includes an MLP encapsulating two attention modules at each layer of the model. Excluding these two structures, ESSformer appears to be a simplified version of TSMixer, which is already effective enough. The results of ablation experiments are also unsatisfactory, as the predictive performance of the ablated models closely resembles that of the non-ablated models. Additionally, in Table 1, we observe that the performance difference between ESSformer and TSMixer is not significant.
2. The reason why randomly partitioning features seems to work could be simply because it involves using a reduced set of features for prediction. The author's perspective is that R-PartA works by diversifying the training set by allowing partial feature interactions, and the model is capable of "overcoming feature dropping" (as discussed in Section 4.3). However, for time series forecasting tasks, the results of univariate time series predicted univariate time series are often superior to multivariate time series predicted univariate time series. Also, the more features are used, the more challenging the prediction becomes, leading to poorer model performance in terms of metrics, so there is no "overcoming feature dropping".
3. In the complexity section, the author does not compare all the experimentally selected baseline models in Table 1, including some Transformer-based models such as PatchTST. This omission is somewhat perplexing.

**Questions:**

I hope the author will address the questions raised in the Weaknesses section in their response.

---

> ### Author Response · Authors · 2023-11-17
> **Response to Reviewer AR89 [1/3]**
>
> Dear Reviewer AR89,
>
> We sincerely thank you for your helpful feedback and insightful comments, especially for acknowledging our presentation and extensive experiments and analyses. In what follows, we address your concerns one by one.
>
> - - -
> About **[Q1]**, we decompose it into three sub-questions and answer them one by one.
>
> **[Q1-1]** ESSformer is not a purely Transformer-based model. \
> **[A1-1]** Our ESSformer is built upon a pure (vanilla) Transformer. There is a typo in the equation (3). We are sorry for the confusion and its corrected version is as follows:
> $$
> \bar{\mathbf{H}}^{(\ell-1)} =\mathbf{H}^{(\ell-1)}+\mathtt{R}\text{-}\mathtt{PartA}(\mathbf{H}^{(\ell-1)},\mathtt{PeriA}(\mathbf{H}^{(\ell-1)})),
> $$
>
> $$
> \mathbf{H}^{(\ell)} =\bar{\mathbf{H}}^{(\ell-1)}+\mathtt{MLP}(\bar{\mathbf{H}}^{(\ell-1)}),
> $$
> where $\mathbf{H}^{(\ell)} \in \mathbb{R}^{N_S \times D \times d_h}$ is the hidden vector of each segment, where $\ell$ is the index of layers, $N_S$ is the number of segments, $D$ is the number of features, and $d_h$ is the hidden size. If we have replaced our sparse attention modules (R-PartA and PeriA) with a vanilla multi-head self-attention (MHSA) of which the attention map is in $\mathbb{R}^{N_SD \times N_SD}$, ESSformer would be exactly the same with vanilla Transformer [1].
>
> **[Q1-2]** Excluding two attention modules, ESSformer appears to be a simplified version of TSMixer. \
> **[A1-2]** ESSformer is totally different from TSMixer in the sense of how to mix temporal and feature information in each layer. To be specific, ESSformer uses sparse attention along temporal and feature axes separately while TSMixer uses MLPs. Also, it is worth noting that the MLP in ESSformer operates on only the embedding axis, not temporal nor feature axes (i.e., $\text{our MLP}: d_h \rightarrow d_h$ where $H\in \mathbb{R}^{N_S\times D\times d_h}$). Therefore, the MLP layer of ESSformer is also different from that of TSMixer. In summary, even without our attention modules, ESSformer is not a simplified version of TSMixer.
>
> **[Q1-3]** The performance gain of ESSformer is not significant compared to the TSMixer and in ablation experiments. \
> **[A1-3]** We respectfully disagree with your opinion about the non-significant empirical improvement. In the M-LTSF benchmark datasets, our performance gain can be regarded as being significant. To further verify that our empirical improvements are significant, we additionally compare ESSformer with concurrent works [4-6] submitted in ICLR 2024 under the same experimental setup. We also provide full ablation results of Table 3 on all datasets. We here report an average of their forecasting performance (in terms of MSE) across all forecasting horizons in seven datasets. Refer to Table 8 of Appendix F and Table 10 of Appendix I for full results. Somewhat surprisingly, ESSformer still outperforms the very recent works with a large margin. Moreover, our ESSformer consistently shows superiority to the ablated one. We think these results demonstrate our empirical significance well. (It is worth noting that our ESSformer not only achieves the best forecasting performance but also reduces the quadratic costs.)
>
> |Paper                          |ETTh1|ETTh2|ETTm1|ETTm2|Weather|Elec.|Traffic|Avg. MSE|Avg. Rank |
> |:-----------------------------|:---:|:---:|:---:|:---:|:-----:|:---:|:-----:|:------:|:--------:|
> |[2] PatchTST                   |0.413|0.331|0.353|0.256|0.226  |0.159|0.391  |0.304   |6.71      |
> |[3] TSMixer                    |0.412|0.355|0.347|0.267|0.225  |0.160|0.408  |0.310   |6.14      |
> |ESSformer (ours)               |**0.392**|0.320|**0.340**|**0.243**|**0.217**  |**0.149**|**0.382**  |**0.292**   |**1.14**      |
> |ESSformer w/o PeriA (ours)     |0.395|0.323|0.346|0.244|**0.217**  |0.151|0.386  |0.294   |2.57      |
> |ESSformer w/o R-PartA (ours)   |0.393|0.321|0.357|0.244|0.222  |0.161|0.395  |0.299   |4.43      |
> |[4] PITS (review score: 6.25)  |0.401|0.332|0.341|0.244|0.225  |0.158|0.400  |0.300   |4.43      |
> |[5] ModernTCN (review score: 7)|0.404|0.322|0.351|0.253|0.224  |0.156|0.396  |0.301   |4.86      |
> |[6] FITS (review score: 6.5)   |0.401|**0.309**|0.356|0.249|0.219  |0.160|0.403  |0.300   |4.71      |

---

> ### Author Response · Authors · 2023-11-17
> **Response to Reviewer AR89 [2/3]**
>
> - - -
> **[Q2-1]** The reason why randomly partitioning features seems to work could be simply because it involves using a reduced set of features for prediction. For time series forecasting tasks, the results of univariate time series predicted univariate time series are often superior to multivariate time series predicted univariate time series. Also, the more features are used, the more challenging the prediction becomes, leading to poorer model performance in terms of metrics. \
> **[A2-1]** Thank you for pointing out the important topic in the time-series forecasting literature. Fundamentally, multivariate forecasting models should be better than univariate ones because the former can utilize more information when forecasting. The recent time-series forecasting literature has also empirically shown that the multivariate models outperform univariate ones [5,7-13]. However, as you mentioned, utilizing too many features does not always lead to the performance improvement because it makes the task challenging too much. Therefore, it is important to consider multiple features adequately for forecasting.
>
> Interestingly, our random partitioning module can cope with the trade-off as shown in Figure 4(b): the simplest and most difficult cases ($S_G=1$ and $S_G=D$, respectively) are worse than the best case ($S_G=20$). To further address your concern, we conduct additional experiments with varying the group size $S_G$ on more datasets (ETTm1, Weather, Electricity, Traffic). The experiments (Figure 9 of Appendix H) shows that (1) the univariate models ($S_G=1$) are not optimal in contrast to your concern, (2) full-multivariate models ($S_G=D$) are also not optimal in accordance with your expectation, and (3) the most diverse case (based on the number of partitions $N_P$) generally can provide the best performance.
>
> **[Q2-2]** There is no "overcoming feature dropping" (in the last paragraph of Section 4.3). \
> **[A2-2]** First, we are sorry for the confusion. We want to clarify that the paragraph aims to claim that ESSformer can deal with missing features (i.e., robust to feature dropping) using the idea of random partitioning. We revise the paragraph to address your concern.

---

> ### Author Response · Authors · 2023-11-17
> **Response to Reviewer AR89 [3/3]**
>
> - - -
> **[Q3]** In the complexity section, the author does not compare all the experimentally selected baseline models in Table 1, including some Transformer-based models such as PatchTST. This omission is somewhat perplexing. \
> **[A3]** We provide theoretical analyses of all baselines in Table 1 about computational costs with an average of performance scores (MSE) and rank across all datasets. $T$ is the input historical time length and $D$ is the number of features. Also, $S=\frac{T}{N_S}$ is the size of each segment in segment-based Transformer, where $N_S$ is the number of segments. This table shows that our ESSformer gives the best forecasting performance with a quite low cost. This table is included in Appendix D.2.
>
> |Method     |Theoretical Complexity                         |Avg. MSE|Avg. Rank|
> |:---------|:---------------------------------------------:|:------:|:-------:|
> |ESSformer  |$\mathcal{O}(D \left(\frac{T}{S}\right)^{1.5})$|**0.292**   |**1.04**    |
> |Crossformer|$\mathcal{O}(D \left(\frac{T}{S}\right)^{2})$  |0.650   | 7.21    |
> |PatchTST   |$\mathcal{O}(D \left(\frac{T}{S}\right)^{2})$  |0.304   | 2.86    |
> |FEDformer  |$\mathcal{O}(T)$                               |0.373   | 6.93    |
> |Pyraformer |$\mathcal{O}(T)$                               |0.888   |10.29    |
> |Informer   |$\mathcal{O}(T \log T)$                        |1.170   |10.43    |
> |TSMixer    |$\mathcal{O}(DT^{2}+D^{2}T)$                   |0.310   | 2.79    |
> |NLinear    |$\mathcal{O}(DT)$                              |0.319   | 4.61    |
> |NLinear-m  |$\mathcal{O}(D^2 T)$                           |N/A     | N/A     |
> |MICN       |$\mathcal{O}(D^2 T)$                           |0.486   | 8.00    |
> |TimesNet   |$\mathcal{O}(DT)$                              |0.387   | 7.25    |
> |DeepTime   |$\mathcal{O}(DT)$                              |0.328   | 4.36    |
>
> [1] Vaswani et al., Attention is All You Need, NeurIPS 2018
> [2] Nie et al., A Time Series is Worth 64 Words: Long-term Forecasting with Transformers, ICLR 2023
> [3] Chen et al., TSMixer: An All-MLP Architecture for Time Series Forecasting, 2023
> [4] Anonymous, Learning to Embed Time Series Patches Independently, submitted to ICLR 2024
> [5] Anonymous, ModernTCN: A Modern Pure Convolution Structure for General Time Series Analysis, submitted to ICLR 2024
> [6] Anonymous, FITS: Modeling Time Series with 10k Parameters, submitted to ICLR 2024
> [7] Lin et al., PETformer: Long-term Time Series Forecasting via Placeholder-enhanced Transformer, 2023
> [8] Gao et al., Client: Cross-variable Linear Integrated Enhanced Transformer for Multivariate Long-Term Time Series Forecasting, 2023
> [9] Xue et al., Make Transformer Great Again for Time Series Forecasting: Channel Aligned Robust Dual Transformer, 2023
> [10] Chen et al., A Joint Time-frequency Domain Transformer for Multivariate Time Series Forecasting, 2023
> [11] Zhang et al., SageFormer: Series-Aware Graph-Enhanced Transformers for Multivariate Time Series Forecasting, 2023
> [12] Yu et al., DSformer: A Double Sampling Transformer for Multivariate Time Series Long-term Prediction, CIKM 2023
> [13] Zhang et al., OneNet: Enhancing Time Series Forecasting Models under Concept Drift by Online Ensembling, NeurIPS 2023

---

> ### Author Response · Authors · 2023-11-22
> **A Gentle Reminder to Reviewer AR89**
>
> Dear Reviewer AR89,
>
> Thanks again for your time and effort in reviewing our paper! As the discussion period is coming to a close, we would like to know if we have resolved your concerns expressed in the original reviews. We remain open to any further feedback and are committed to making additional improvements if needed. If you find that these concerns have been resolved, we would be grateful if you would consider reflecting this in your rating of our paper : )
>
> Best regards,
> Authors.

---

### Official Review · Reviewer_EwZF · 2023-11-01

**Soundness:** 3 good
**Presentation:** 3 good
**Contribution:** 2 fair
**Rating:** 6
**Confidence:** 4

**Summary:**

This is paper presents a new Transformer variant for multivariate long-term time series forecasting. In particular, it proposes to decompose tokenized long time series into blocks over both temporal and feature dimensions. For temporal contextualization, dense attention within each block is followed by dilated attention across blocks; for feature contextualization, the partitioning is done randomly and dense attention is applied within each random group only. In this way, the complexity of self-attention is reduced while the random partitioning in features introduces implicit data augmentation to boost the performance.

**Strengths:**

1. The paper is well-written with clarity in the presentation of the well-motivated main ideas;
2. The empirical studies are extensive with strong results.
3. Abundant analyses are provided to support the effectiveness of the proposed methods.

**Weaknesses:**

1. The novelty of such a Transformer variant is rather limited given the mediocre empirical improvement.
2. The name of the proposed method is somewhat misleading. For example, "periodic and random sparsity" in the title is not accurate as sparsity only applies to temporal dimension and random only applies to feature dimension, while these highlights are placed side-by-side. Moreover, the term "periodic" is confusing since it has nothing to do with the intrinsic periodicity of time series, as the block size $P$ is empirically selected.
3. The theoretical analysis on complexity needs to be elaborated.

**Questions:**

1. Why is the complexity of R-PartA reduced to be $\mathcal{O}(DS_G)$?

---

> ### Author Response · Authors · 2023-11-17
> **Response to Reviewer EwZF [1/2]**
>
> Dear Reviewer EwZF,
>
> We sincerely thank you for your helpful feedback and insightful comments, especially for acknowledging our presentation and extensive experiments and analyses. In what follows, we address your concerns one by one.
>
> - - -
> **[Q1]** The novelty of such a Transformer variant is rather limited given the mediocre empirical improvement. \
> **[A1]** We respectfully disagree with your opinion about the mediocre empirical improvement and the limited novelty. To further verify that our empirical improvements are significant, we additionally compare ESSformer with concurrent works [2-4] submitted in ICLR 2024 under the same experimental setup. We here report an average of their forecasting performance (in terms of MSE) across all forecasting horizons. Please check Appendix F for the full results. Somewhat surprisingly, ESSformer still outperforms the very recent works by a large margin. We think these results demonstrate our empirical significance well.
>
> |Paper                          |ETTh1|ETTh2|ETTm1|ETTm2|Weather|Elec.|Traffic|Avg. MSE|Avg. Rank |
> |:-----------------------------|:---:|:---:|:---:|:---:|:-----:|:---:|:-----:|:------:|:--------:|
> |[1] PatchTST                   |0.413|0.331|0.353|0.256|0.226  |0.159|0.391  |0.304   |4.14      |
> |ESSformer (ours)               |**0.392**|0.320|**0.340**|**0.243**|**0.217** |**0.149**|**0.382**  |**0.292**   |**1.14**      |
> |[2] PITS (review score: 6.25)  |0.401|0.332|0.341|0.244|0.225  |0.158|0.400  |0.300   |3.14      |
> |[3] ModernTCN (review score: 7)|0.404|0.322|0.351|0.253|0.224  |0.156|0.396  |0.301   |3.14      |
> |[4] FITS (review score: 6.5)   |0.401|**0.309**|0.356|0.249|0.219  |0.160|0.403  |0.300   |3.29      |
>
> Furthermore, as highlighted by Reviewer 6AJS and h291, we strongly believe that our architecture is novel from the following perspectives:
> 1. Our inter-feature attention based on **random partitioning** can provide a new angle of how to deal with multiple features in the multivariate long-term time-series forecasting  (M-LTSF) problem. Compared to ours, the prior works have ignored a correlation between features using an univariate architecture [1,9,10] or have considered all relationships between features [5,6,7,8], which simplifies or complexifies the M-LTSF problem too much, respectively. In addition, the random partitioning technique enables to improve forecasting performance via test-time ensemble.
> 2. Our periodic temporal attention is built upon **the concrete observation of attention maps in segment-based Transformer**. As the segmentation-based Transformer have recently emerged (e.g., PatchTST [1] in ICLR 2023, PITS [2] in ICLR 2024 submission) but their attention patterns are not yet explored, we believe our observation can provide an useful insight into architecture designs.
> 3. To the best of our knowledge, our work is the first to consider **sparse attention for both temporal and feature correlations** in the time-series forecasting literature. We believe ESSformer can be a new strong baseline in M-LTSF benchmarks and accelerate architectural research for time-series forecasting.
>
>
> - - -
> **[Q2]** The name of the proposed method is somewhat misleading. \
> **[A2]** Thank you for pointing out the confusing terms. To address your concern, we will change our title to **"Dilated Temporal Attention and Random Feature Partitioning for Efficient Forecasting"**. Furthermore, we will replace the “periodic” term with “dilated”. (But to avoid some confusion for other reviewers, we maintain the original term during this rebuttal period.) If you have a better suggestion for the title and the terms, please let us know. We are open to such suggestions.

---

> ### Author Response · Authors · 2023-11-17
> **Response to Reviewer EwZF [2/2]**
>
> - - -
> **[Q3]** The theoretical analysis on computation costs has to be elaborated. \
> **[A3]** As you suggested, we include the theoretical analysis on computation costs in Appendix D.1. We also briefly elaborate on the analysis below.
>
> **Computation cost of block-diagonal self-attention.** Because our two attention modules can be formulated by block-diagonal self-attention, we first derive the computation cost of block-diagonal self-attention. Let's assume that $B$ is the size of each block and $N$ is the total number of tokens. Then, the attention cost of each block is $\mathcal{O}(B^2)$ and the number of blocks is $N/B$. Therefore, the computation cost of block-diagonal self-attention is $\mathcal{O}(NB)$.
>
> **Computation cost of PeriA.** PeriA can be formulated by two block-diagonal self-attention modules. Given a period of $P$ and the number of segments (tokens) $N_S$, the block sizes of two modules are $B_1=P$ and $B_2=N_S/P$. As such, the cost of PeriA can be written as $\mathcal{O}(N_S P+ N_S^2/P)$. If we set $P$ to $\sqrt{N_S}$ for efficiency, the final cost becomes $\mathcal{O}(N_S^{1.5})$.
>
> **Computation cost of R-PartA.** R-PartA can be formulated by one block-diagonal self-attention module. Given a group size of $S_G$ and the number of entire features $D$, the block size is $B=S_G$ and the final cost of R-PartA is $\mathcal{O}(D S_G)$.
>
> [1] Nie et al., A Time Series is Worth 64 Words: Long-term Forecasting with Transformers, ICLR 2023
> [2] Anonymous, Learning to Embed Time Series Patches Independently, submitted to ICLR
> 2024
> [3] Anonymous, ModernTCN: A Modern Pure Convolution Structure for General Time Series Analysis, submitted to ICLR 2024
> [4] Anonymous, FITS: Modeling Time Series with 10k Parameters, submitted to ICLR 2024
> [5] Zhang et al., Crossformer: Transformer Utilizing Cross-Dimension Dependency for Multivariate Time Series Forecasting, ICLR 2023
> [6] Ni et al., BasisFormer: Attention-based Time Series Forecasting with Learnable and Interpretable Basis, NeurIPS 2023
> [7] Wang et al., MICN: Multi-scale Local and Global Context Modeling for Long-term Series Forecasting, ICLR 2023
> [8] Wu et al., TimesNet: Temporal 2D-Variation Modeling for General Time Series Analysis, ICLR 2023
> [9] Zeng et al., Are Transformers Effective for Time Series Forecasting?, AAAI 2023
> [10] Orenshkin et al., N-BEATS: Neural basis expansion analysis for interpretable time series forecasting, ICLR 2020

---

> ### Author Response · Authors · 2023-11-22
> **A Gentle Reminder to Reviewer EwZF**
>
> Dear Reviewer EwZF,
>
> Thanks again for your time and effort in reviewing our paper! As the discussion period is coming to a close, we would like to know if we have resolved your concerns expressed in the original reviews. We remain open to any further feedback and are committed to making additional improvements if needed. If you find that these concerns have been resolved, we would be grateful if you would consider reflecting this in your rating of our paper : )
>
> Best regards,
> Authors.

---

> > ### Comment · Reviewer_EwZF · 2023-11-22
> > **Response to authors**
> >
> > Thanks for your detailed responses. I believe the authors have addressed all my concerns. I am happy to update my rating.

---

### Official Review · Reviewer_6AJS · 2023-11-04

**Soundness:** 3 good
**Presentation:** 4 excellent
**Contribution:** 3 good
**Rating:** 6
**Confidence:** 3

**Summary:**

The paper proposes a transformer approach for multivariate timeseries forecasting. To improve the efficiency for segment transformers authors propose two new self-attention modules. First module applies block-diagonal and stride dilated attention to capture temporal relationships. Second module partitions features into disjoint groups and applies attention within each group. Empirically, authors show that the combination of the two modules leads to better performance and higher efficiency.

**Strengths:**

I found the paper to be well written and easy to follow. The two introduced attention blocks are interesting and while numerous papers have been published on attention, I believe this work is novel. Authors provide extensive empirical evaluation showing superior performance over a number of leading baselines. Furthermore, extensive ablation study is conducted demonstrating the contribution of each introduced module and the efficiency gains in computation.

**Weaknesses:**

The R-PartA introduces randomness into the grouping of the features. To overcome this during inference authors run multiple passes through the model and ensemble the results. This can create an unfair advantage as ensembles nearly always improve performance and often significantly, Figure 5 further validates it. I think the results reported in Table 1 should not be ensembled and I suspect that this would significantly reduce the performance improvements.

**Questions:**

What is the average rank of ESSformer (Table 1) if the ensembling is not used?

---

> ### Author Response · Authors · 2023-11-17
> **Response to Reviewer 6AJS**
>
> Dear Reviewer 6AJS,
>
> We sincerely thank you for your helpful feedback and insightful comments, especially for acknowledging the novelty in our work with abundant experiments to prove the effectiveness of ESSformer. In what follows, we address your concerns one by one.
>
> - - -
> **[Q1]** The ensemble method may cause an unfair comparison issue as such an ensemble technique often significantly improves performance. What is the average rank of ESSformer (Table 1) if the ensembling is not used? \
> **[A1]** First, we want to emphasize that ESSformer still remains to be state-of-the-art (the average rank = 1.107) even without our test-time ensemble technique. In addition, we believe our comparison is fair because all the baselines do not support such a test-time ensemble without our component, R-PartA. Namely, our ensemble technique is also our own methodological contribution, not a simple application of the conventional ensemble. To be specific, our idea is to utilize multiple partitions during inference using a single model, but the conventional ensemble technique is based on multiple (independently-trained) models. We will replace the term “ensemble” with “test-time ensemble” in our manuscript to mitigate your concern.
>
> For your information, we here provide the summary of average rank results without our test-time ensemble technique (i.e., $N_E=1$). Please check Appendix G for full experimental results. As reported below and in Appendix G, ESSformer without ensemble still consistently outperforms the baseline methods by a significant margin. This shows that our gain mainly originates from our architectural improvements during the training phase.
>
>
> |ESSformer w/o ensemble|PatchTST|FEDformer|TSMixer|DeepTime|
> |:--------------------:|:------:|:-------:|:-----:|:------:|
> |**1.107**                 |2.679   |4.821    |2.500  |3.786   |

---

> ### Author Response · Authors · 2023-11-22
> **A Gentle Reminder to Reviewer 6AJS**
>
> Dear Reviewer 6AJS,
>
> Thanks again for your time and effort in reviewing our paper! As the discussion period is coming to a close, we would like to know if we have resolved your concerns expressed in the original reviews. We remain open to any further feedback and are committed to making additional improvements if needed. If you find that these concerns have been resolved, we would be grateful if you would consider reflecting this in your rating of our paper : )
>
> Best regards,
> Authors.

---

### Author Response · Authors · 2023-11-17
**Global Response (Summary of Revisions)**

Dear reviewers and AC,

We sincerely appreciate your valuable time and effort spent reviewing our manuscript.

As reviewers highlighted, our work proposes an interesting/novel method (Reviewer 6AJS, h291) with strong empirical results (Reviewer 6AJS, EwZF, h291), provides extensive analyses (Reviewer 6AJS, EwZF, AR89), and provides well-written/easy-to-follow presentation (ALL Reviewers).

We appreciate your constructive comments on our manuscript. In response to the comments, we have carefully revised and enhanced the manuscript with the following additional discussions and experiments:

- Clarification about some misleading terms or sentences.
  - Title &rarr;  Dilated Temporal Attention and Random Feature Partitioning for Efficient Forecasting.
  - Ensemble method &rarr; Test-time ensemble method.
  - Periodic Attention (PeriA) &rarr; Dilated Attention (DilA)) *[We will change it after this rebuttal period, to avoid reviewers’ confusions]*.
  - overcome feature dropping (in Section 4.3) &rarr; deal with the situation where some features are missing.
- Correction about our equations (Equation (3)~(4)).
- Elaboration on the computation costs of ESSformer (Appendix D.1).
- Additional analysis on computational costs of ESSformer and baselines (Table 6 and Appendix D.2).
- Additional experimental comparison with very recent concurrent works (Table 8 and  Appendix F).
- Additional experiments about the forecasting performance of ESSformer without the test-time ensemble method (Table 9 and Appendix G).
- Additional analyses on $S_G$ and $N_E$ (Figure 9, Figure 10, and Appendix H).
- Additional ablation studies (Table 10 in Appendix I).

These updates are temporarily highlighted in “red” for your convenience to check.

We hope our response and revision sincerely address all the reviewers’ concerns.

Thank you very much.

Best regards,
Authors.

---

### Meta-Review · Area_Chair_ARqZ · 2023-12-07

**Metareview:**

The paper introduces Efficient Segment-based Sparse Transformer (ESSformer), which incorporates two sparse attention modules, Periodic Attention and Random-Partition Attention, to efficiently capture temporal and inter-feature dependencies in multivariate long-term time-series forecasting, outperforming various baselines while reducing quadratic complexity. This is a clear borderline paper. After carefully reading the reviews and the authors' rebuttal, I am inclined not to recommend its acceptance. The reasons are in line with reviewer EwZF's judgment: "The novelty of such a Transformer variant is rather limited given the mediocre empirical improvement." The two improvements proposed in this paper are relatively incremental in terms of their approach, and the experimental results do not show significant gains or clear reasons for the improvement. Additionally, this paper focuses on a relatively specific problem, making it difficult to generalize its methods to more impactful efficient transformer architectures. Its future applications might be limited.

**Justification For Why Not Higher Score:**

n/a

**Justification For Why Not Lower Score:**

n/a

---

### Decision · Program_Chairs · 2024-01-16

Reject